# Towards data-driven sign language interpreting Virtual Assistant

## ABSTRACT

Sign Languages (SL) are a form of communication in the visual-gestural modality, and are full-fledged natural languages. Recent years have seen the increase in the use of virtual avatars as assistants for sign language users. Research into sign language recognition has demonstrated promising potential for the improvement of the communication with deaf people. However, the area of sign language synthesis is still in its infancy. This explains the underdevelopment of virtual intelligent signing systems, which could bridge the communication with the deaf and make it more favorable. In addition, existing models are often restricted to manually written rules and require expert knowledge, while data-driven approach could provide the better solution.

In this paper, we present a user study on the evaluation of the data-driven Virtual Assistant that performs manual gestures for Kazakh-Russian Sign Language using sign sequences. The study sets out to answer three research questions concerning the users' perceptions and feedback on the performance of the four signing avatars, namely two data-driven avatars, one motion capture animation avatar and a human sign interpreter. The results of the questionnaire suggest that while the signing avatars generally perform well, they could not outperform the human agent in terms of naturalness and likeability. Hence, a further study might include the improvements necessary to increase the naturalness of the manual gestures.

## KEYWORDS

sign language, virtual assistant, generation, HRI

**ACM Reference Format:**
. 2021. Towards data-driven sign language interpreting Virtual Assistant. In *Proceedings of ACM Conference (Conference'17)*. ACM, New York, NY, USA, 6 pages. https://doi.org/10.1145/nnnnnnn.nnnnnnn

## 1 INTRODUCTION

The presence of intelligent virtual assistants (IVA) in our day-to-day lives is not at all a new phenomenon. They have become an integral part of human-agent interaction, providing a wide range of functionalities, including the establishment of contact with humans through verbal and non-verbal communication channels [30].

While a majority of existing work focuses on spoken/written languages, another large class of languages exists that uses the visual-gestural modality for interaction, namely sign languages.

*Conference'17, July 2017, Washington, DC, USA*
© 2021 Association for Computing Machinery.
ACM ISBN 978-x-xxxx-xxxx-x/YY/MM...$15.00
https://doi.org/10.1145/nnnnnnn.nnnnnnn

Sign languages are full-fledged natural languages used by deaf communities around the world. Similar to spoken languages, different sign languages exist in different countries and regions, and they vary in phonology, morphology, lexicon, semantics, syntax and pragmatics [27]. A majority of existing works that focused on the synthesis of spoken/written natural languages inspired the sign language synthesis, resulting in integration of the existing techniques to animate sign languages [30].

Despite the common misconception, sign languages are not articulated solely by the hands [29]. In fact, both manual and non-manual gestures are crucial components of sign languages [26] [33]. More precisely, the former includes gesture features such as those related to hands (e.g., hand configuration and motion trajectory of hands), whereas the latter involves head and body movements and movements of facial muscles (e.g., facial expressions, gaze direction, lip pattern, and head and body postures) to convey information [26] [29].

In a manner resembling humans, IVAs present a range of advantages for the communication of the deaf, offering synthesis and interpretation from a spoken/written language to a sign language and vice versa [5]. Compared to videos of human sign language interpreters, computer-supported sign language systems are sought after due to their flexibility [9]. Delorme et al. [9] highlight the ability of a signing avatar to produce various sentences from a database of isolated signs as one of its advantages. A considerable literature has grown up around the theme of sign language synthesis, giving insight into various methods and frameworks for modeling sign language recognition and generation systems [30] [31] [21] [17].

Most of the existing models for sign language synthesis are based on rules [36] [24] [12] [35]. While rule-based algorithms perform well, they are often costly, time-consuming and bound up to expert knowledge. In addition, rule-based models are often limited to certain pre-defined types of gestures [20], and therefore might fail to produce both the manual and non-manual parameters of the sign language.

In contrast, data-driven systems learn from data without the need of expert knowledge [20]. Creating an automatic sign language generation for virtual avatars has gained importance with the rise of data-driven systems (see Kipp et al. [19]). Earlier works relied on parametric and geometric approaches [9], while most recently Kipp et al. [19] presented a fully synthesized model and Gibet et al. [11] and Ebling and Huenerfauth [10] proposed semi-automatically synthesized models for sign language generation using small corpora of manual gesture data. It is noteworthy that these models were generally designed for the well-researched sign languages such as American Sign Language (ASL), German Sign Language (DGS) [10], and French Sign Language (LSF) [11], compared to the relatively less explored sign languages [5].

The goal of this work is to create a data-driven avatar for sign language generation and evaluate its performance in a user study with participants who have a command of Kazakh-Russian Sign

Language (K-RSL). The evaluation is based on the standard questionnaire (Godspeed [4]) encompassing certain sets of evaluation metrics designed for the general use in the human-robot interaction (HRI) research.

We begin by training our IVA/VA mock avatar on the dataset of recorded videos where people perform K-RSL sentence sequences. To estimate human poses, we utilize OpenPose Cao et al. [6]. The obtained 3D movement predictions are then converted to a Visual Molecular Dynamics (VMD) [13] format file. Consequently, VMD files are uploaded to Unity3D, where they program motions of virtual characters. The resulting video of virtual avatar's performance of the K-RSL sentences is watched by 18 participants recruited for the user study for the purpose of acquiring their perceptions and feedback on the presented avatars.

## 2 OBJECTIVE

This study addresses the Signing VAs' performance and perception by deaf signers, and answers the following research questions:

- How is the concept of data-driven Signing Virtual Assistant perceived by deaf respondents? Performance feedback.

- Comparison of data-driven and manually programmed signing virtual assistants.

- What can be improved according to deaf feedback?

## 3 UNSUCCESSFUL ATTEMPTS

To begin with, we surveyed the state-of-the-art models and methods designed to capture gestures and movements for sign language generation so as to integrate them as subparts into a fully-fledged Signing Virtual Avatar.

### 3.1 Monocular Total Capture.

A proper and understandable signing requires accurate finger mapping, facial expressions, head and body tilt. The first approach we came across was Monocular Total Capture (MTC) [37]. MTC is the first method that captures the 3D total motion of humans from monocular images or videos and reconstructs the whole body pose by a 3D deformable mesh model. Authors use representation called 3D Part Orientation Fields in the first stage. In the second stage, image measurements produced by CNN are taken and then fitting deformable the human mesh model on these measurements. After this, motion jitters reducing. To train CNNs, the authors involved 40 subjects who performed different motions of body, hands, and face. We tested it on videos taken from our dataset, which has been collected almost completely and will be presented further. This dataset is supposed to be a subpart of Kazakh-Russian Sign Language Corpus together with other subparts [18] [26] [14].

As can be seen in Figure 1 a, Monocular Total Capture (MTC) performs perfectly for the hands: the reconstruction of finger configurations is highly accurate, except for cases of slight overlapping, which are normally insignificant. Unfortunately, face reconstruction that expresses the mouthing and facial expressions could not be obtained. This complicates the recognition of the sentence either as a question or a statement. Additionally, sentiment recognition in general turns out to be tricky.

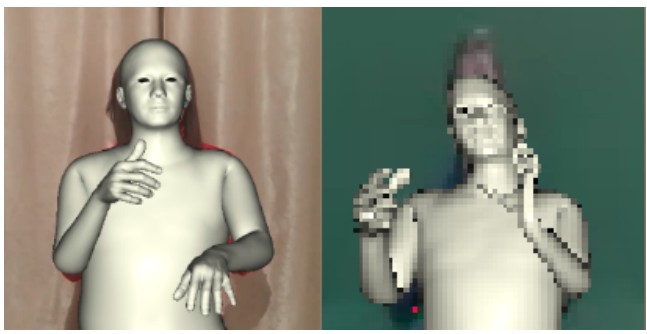

**Figure 1: a) Performance of Monocular Total Capture approach for videos containing Kazakh-Russian Sign Language sentences, b) Performance of Monocular Total Capture approach for a fake generated video. Video was generated by MoCoGAN approach trained with videos taken from the K-RSL dataset.**

### 3.2 MoCoGan

We have also attempted to test MTC's performance on fake-generated videos. We trained Motion and Content Generative Adversarial Network (MoCoGAN) [34] on our videos.

As a generative adversarial framework for fake video generation, MoCoGAN generates video by vectors with two subparts for motion and content, where the 'content' part is fixed and the 'motion' part is stochastic. While the content is for objects that appeared in a video, 'motion' shows the dynamics of these objects.

The architecture of MoCoGAN contains 4 RNNs: Motion subspace $Z_m = R_m$ as one-layer GRU, image generator $G_i$, image discriminator $D_i$, and video discriminator $D_v$. $D_i$ criticizes $G_i$ based on individual images (it can determine if an image is from real videos), and $D_v$ criticizes $G_i$ based on generated videos. Experiments showed that MoCoGAN can generate videos of the same object with different motions or of different objects performing the same motion. That is why we generated fake videos based on videos from our dataset and ran Monocular Total Capture on them.

Monocular Total Capture performed relatively well as it could reconstruct the fingers, considering that MoCoGAN produced 96x96 pixel fake videos (see Figure 1 b). Despite promising results on hand reconstruction, it still fails to provide proper facial expressions, concomitantly lacking human-likeness.

## 4 METHODOLOGY

Initially we intended to test the performance on an NAO avatar, with the intention to transfer obtained coordinates to a real NAO robot available in the lab in the foreseeable future. For this reason, we chose only signs with configurations involving only the open palm (with all fingers selected), as the robot can only perform such configurations. However, the NAO avatar does not have enough DOFs to even express these hand configurations. At this stage, we tried free characters from the Unity Asset Store [1] to express signing sequences. We aimed at summarizing user experience and evaluation of signing gained during the experiment sessions when participants watched videos of four avatars performing sign language sequences. To formalize the mock of a signing avatar we

present it as a concept, we used implementations that utilize several tools such as OpenPose [6] and Unity3D. These implementations include Autotrace [2] and OpenMMD [28]. We tried to check the performance of the first one. It consists of the four steps (see Figure 2) described further.

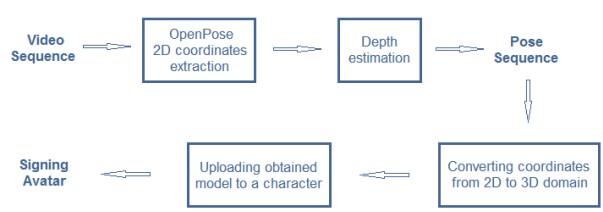

Figure 2: Pipeline of tested approach

## 4.1 OpenPose (Extraction of 2D human body coordinates from videos)

OpenPose is a tool developed by Carnegie Melon University researchers aimed at Human Pose estimation in 2D. Generally, it finds and localizes anatomical keypoints (see Figure 5). It simultaneously utilizes two techniques: Confidence Maps for body parts detection and Part Affinity Fields to associate body parts if they belong to the same human and then match them to get keypoints representation. We utilized OpenPose to extract signers' full-body coordinates.

## 4.2 Depth estimation (Mannequinchallenge-vmd)

A video's depth estimation is implemented as the second step. Authors of the method [22] present a data-driven approach that aimed at depth prediction for videos where people and a monocular camera move freely. For this, they collected a dataset called Mannequin-Challenge [3] and performed supervised learning to train their depth prediction model. They used the Multi-View Stereo (MVS) [32] approach for depth generated and then applied regression. This step extracts human depth regions from the videos, which helps to segment human region and increase accuracy of separate person's keypoints extraction.

## 4.3 3D-pose-baseline to VMD (Converting 2D coordinates into 3D)

There are several considerably similar implementations of the approach described and presented in [25]. This approach provides proper and accurate conversion of human body coordinates from 2D videos into the 3D domain. Authors claim that their method outperforms the other 2D to 3D shifting techniques by almost 30% tested on the Human3.6M dataset [15] (see Figure 3). Also, according to the authors, they use a simple six-layer architecture. Thus, we leveraged the pre-trained models proposed by [25] and tested on our sign language videos (see Figure 4).

To the best of our knowledge, one of the implementations also includes an adversarial subpart (GAN), since the prediction of the

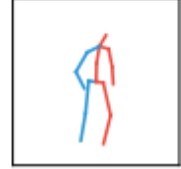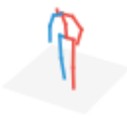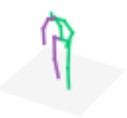

Figure 3: Approach performance on Human3.6M [15] test set. From left to right: 2D observation, 3D ground truth, 3D predictions. Image taken from [25].

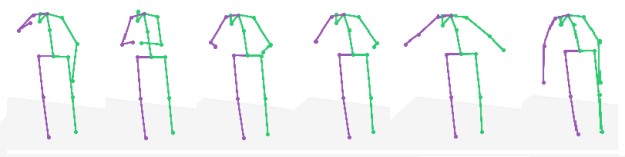

Figure 4: 3D prediction of a Kazakh-Russian sign language sequence.

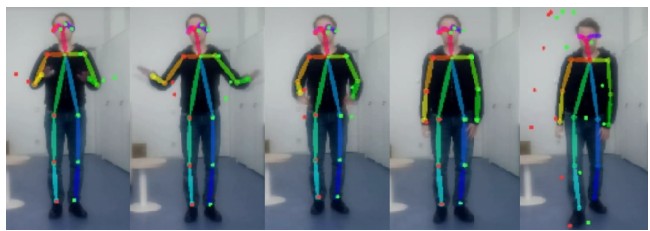

Figure 5: 3D prediction keypoints move earlier than actual body parts.

3D pose precedes the actual movements of the person/signer in the resulting videos (see Figure 5). It is noticeable that red and green body key points of the 3D prediction are outpacing the movements of the body parts.

## 4.4 3D coordinates into VMD model

At this stage, the obtained coordinates of the 3D movement prediction were converted to a Visual Molecular Dynamics (VMD) [13] format file. VMD was primarily designed for computational biophysics studies to make possible modeling of biological systems, namely biological macromolecules such as carbohydrates, proteins, nucleic acids, and lipids. Nowadays it is widely used for 3D visualizations and representations in general.

Once we obtain a VMD model, we upload it into a Unity3D project and link it to the free humanoid characters we chose (see Figure 6). There are screenshots taken when standard free avatars from the Unity Asset Store performing common sentences from the general K-RSL domain.

## 5 USER STUDY

We recruited 18 people and asked them to take our survey. Our online survey consists of a mixture of open and closed questions and questions measured by the Likert scale [16]. The participants were

Figure 6: Four types of avatars: two data-driven ones, one manually programmed one and a human.

provided with videos of four avatars performing signing sequences and were tasked to evaluate its performance by the proposed criteria. Our task is closer to the areas of HRI, HCI and social robots acceptability.

We rely on the use of Likert scale-based questionnaires because of their simplicity and comprehensibility as well as time-efficiency compared to open questions. Our questionnaire is based on the Godspeed [4] questionnaire generally used for human-robot interaction studies. We also formulated and added several new questions since they focused on previously unmentioned situations related to signing performance by IVA as authors of Robotic Social Attributes Scale (RoSAS) [7] done.

There were 10 Likert-scale questions from the Godspeed questionnaire, 11 additional Likert-scaled questions, and four yes/no questions.

The consent form and all the instructions and questions were translated to K-RSL, filmed as short videos and presented to participants during the experiment. Participants received promised monetary compensations for their time and contribution.

## 5.1 Background information

In the beginning, we collected demographic information about our participants along with the information on their level of proficiency in sign language and experience of using it. The questions were designed so as to acquire background information and distinguish between different groups based on their everyday usage of the sign language.

## 5.2 Participants

In total, 18 respondents were involved in the study: 12 deaf participants and 6 hearing interpreters, aged from 18 to 57 (mean age - 33), with the gender distribution of 4 male and 14 female participants. Two participants were from Russia, Yakutsk and graduated from the same school (RSL and K-RSL are very close since both of them originated from the same signing system that was used within the former USSR). The other respondents were from Kazakhstan (Nur-Sultan, Petropavlovsk, Karagandy). Respondents currently located in Nur-Sultan mostly came from different cities and studied in different special education schools. Concerning the education levels, the majority of the participants holds a completed college degree, while only four participants hold a bachelor degree (including one

Table 1: Participants

| Gender | Age | Location | Education | Usage of SL |
|--------|-----|----------|-----------|-------------|
| M | 36 | Nur-Sultan | 9th grade | Deaf |
| F | 37 | Nur-Sultan | College | Interpreter |
| F | 18 | Petropavlovsk | College | Deaf |
| F | 28 | Nur-Sultan | Bachelor | Interpreter |
| M | 33 | Nur-Sultan | College | Deaf |
| F | 20 | Nur-Sultan | Bachelor | Interpreter |
| F | 30 | Nur-Sultan | College | Deaf |
| M | 38 | Karagandy | 11th grade | Deaf |
| F | 35 | Yakutsk | College | Deaf |
| F | 30 | Nur-Sultan | College | Deaf |
| F | 31 | Jaksy | College | Deaf |
| F | 37 | Nur-Sultan | Bachelor | Interpreter |
| F | 21 | Nur-Sultan | College | Interpreter |
| F | 30 | Karagandy | College | Deaf |
| F | 43 | Nur-Sultan | College | Interpreter |
| F | 28 | Petropavlovsk | College | Deaf |
| M | 37 | Yakutsk | Bachelor | Deaf |
| F | 57 | Nur-Sultan | College | Deaf |

deaf participant) and the rest vary from the completed upper high school to high school grades (i.e., 9th grade and 11th grade).

## 5.3 Four avatars

We aimed to understand user perception of two implemented avatars in comparison to manually programmed avatar and a human who is new to sign language. In the study, participants were asked to watch four videos with four avatars (see Figure 6) and answer questions about each avatar. Two of them were our proposed data-driven avatars: the woman in a white blouse and the man wearing a black vest. These two avatars performed sign language phrases that contained signs with an open palm configuration only. Avatar 3 was a manually programmed avatar from [8], [23]. Avatar 3 was created in the laboratory at Queens College of the City University of New York for CUNY ASL Motion-Capture Corpus. This project aimed at the collection of digital 3D body movement and hand-shape data. They use motion capture equipment (sensory gloves) to extract

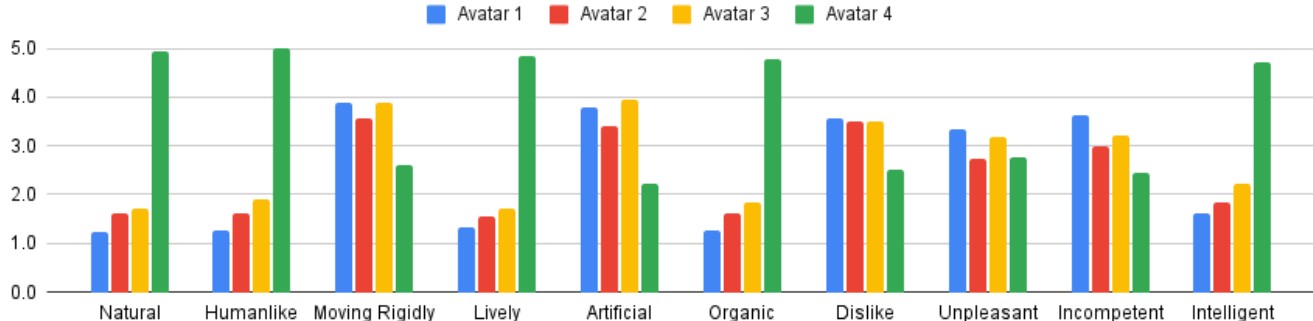

**Figure 7: Average ratings for each question comparing four avatars**

data from the motions of native signers. Since it was originally designed for American Sign Language (ASL), we could find only one of the signing output demo videos that contain open palm signs that also have meaning in K-RSL. We cut it to a short video that was presented to participants. Avatar 4 is a human who is new to sign language and simply repeated a sentence in front of the camera following a real interpreter. We asked Avatar 4 to do so ensure that participants would watch videos closely and to check will some of them notice the lack of sign language experience or not. An online questionnaire consisted of five sections: four sections were used to evaluate each avatar after watching each video using questions from the Godspeed questionnaire. Two data-driven avatars were additionally asked about. We used counterbalancing to swap avatars among each other to avoid ordering effect.

To this end, each avatar expressed one sentence only: Avatar 1 expressed the sign sequence "Nothing new", Avatar 2 performed "Hello" sign twice, Avatar 3 showed "I will stop", while Avatar 4 performed "I like fish". We tried to provide short sequences roughly equivalent in complexity. Participants could watch videos several times.

We conducted a series of Friedman tests to understand if there are significant differences between avatars for each measure. Table 2 displays the results with significant differences presented in bold. For example, we found significant differences in the ratings of Humanlikeness of the avatars: $\chi^2(3) = 39.281; p < 0.001$. with Avatar 1's rating being 1.58, Avatar 2's rating - 1.62, Avatar 3's rating - 1.91, and Avatar 4's rating - 5 (see Figure 7). Pairwise comparison revealed that Avatar 4 was rated significantly higher than other three avatars. Differences between pairs of other avatars were not significant.

Similarly, Table 2 demonstrates that we found significant differences for almost all ratings suggesting that a human was rated as significantly more natural, more lively, more lifelike, more organic as well as more intelligent. These findings suggest that our data-driven avatars need significant improvements to reach the ratings of a human avatar. Interestingly, people did not give significantly different ratings for Moving Elegantly - Moving Rigidly, Competent - Uncompetent, Like-Dislike and Pleasant - Unpleasant between four avatars. This could suggest that our participants generally had mixed feelings towards the appearances of all avatars and perceived them as moderately pleasant.

**Table 2: Friedman test results. Significant findings are in bold.**

| Measurement | Friedman test output |
| --- | --- |
| **Fake - Natural** | $\chi^2(3) = 43.795; p = 0.000.$ |
| **Machinelike - Humanlike** | $\chi^2(3) = 39.281; p = 0.000.$ |
| Moving elegantly - Moving rigidly | $\chi^2(3) = 6.614; p = 0.085.$ |
| **Stagnant - Lively** | $\chi^2(3) = 40.452; p = 0.000.$ |
| **Lifelike - Artificial** | $\chi^2(3) = 8.955; p = 0.03.$ |
| **Mechanical - Organic** | $\chi^2(3) = 42.022; p = 0.000.$ |
| Like-Dislike | $\chi^2(3) = 6.060; p = 0.109.$ |
| Competent - Incompetent | $\chi^2(3) = 6.944; p = 0.074.$ |
| Pleasant - Unpleasant | $\chi^2(3) = 3.358 p = 0.340.$ |
| **Unintelligent - Intelligent** | $\chi^2(3) = 30.163; p = 0.000$ |

## 6 DISCUSSION

One of the most valuable results is that 13 out of 18 participants correctly understood Avatar 2. It could be biased by the fact that that sign was quite easy in comparison to other phrases.

We would like to refer to non-significant differences between data-driven avatars and a manually coded one: Avatar 3 received slightly better ratings but it was never significantly different. We believe that this is a promising finding for our data-driven avatars as they were generated in a completely autonomous manner with multiple limitations, such as an absence of face and fingers movements. Even though we deliberately selected signs that did not require fingers and face movements, our data-driven avatars need further work to avoid this major shortcoming. One of the participants after the experiment mentioned that despite the fact that Avatar 3 performed finger articulations well, hand movements were very fast while the body and head did not move, which was unnatural and probably led to ratings being low for that avatar type.

## 7 CONCLUSIONS AND FUTURE WORK

Although some promising results showed that one of our data-driven avatars (Avatar 2) could deliver its message and performed understandable signing for participants, there is still room for improvement. Respondents' feedback indicates that they need accurate finger articulations, emotions, and mouthing should add for

easier understanding and proper sign language delivery by avatars. This implies that the balance between manual and non-manual features of sign languages is crucial.

The overall results suggest that participants are quite optimistic about the future capabilities of signing IVA technology. That is why we need to improve the performance by adding precise reconstruction for fingers accompanying relevant facial expressions.

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
