# OpenReview forum: "Towards data-driven sign language interpreting virtual assistant"
_ACM.org/ICMI/2021/Workshop/GENEA — Reject_

### Official Review · Reviewer_xaNg · 2021-07-16
**Towards data-driven sign language interpreting virtual assistant**

**Rating:** 4
**Confidence:** 5

**Review:**

This paper presents an evaluation of motion-captured and manually programmed signing avatars, in comparison to a real human performing sign language. The authors are making good progress to assess what is available in terms of open-source solutions, getting familiar with virtual human animation, and investigating their use specifically for Russian sign-language. They also perform a user study to assess the effectiveness of different approaches.

However, the paper itself has some issues. Firstly, it is not totally clear to me which method is used for each of the avatar conditions in the user study. I would suggest to put this information in the user study section, or try to link the user study with the implementation part more. The user study itself is too simplistic with a lot of confounding information which makes it difficult to draw any meaningful conclusions. Here are some suggestions for the authors to improve it:

1. The avatars in the study all look different, some are male and some are female and some have 3D shading, one has flat-shading, and one is a real human. There has been a lot of research showing the different aspects that can impact appeal, which should be kept consistent for a fair comparison. See for example:

Wisessing et al.  showed that 3D vs 2D shading has an effect on appeal: 'Perception of lighting and shading for animated virtual characters' and that even brightness and shadow of the light source has a noticeable effect on appeal: 'Enlighten Me: Importance of brightness and shadow for character emotion and appeal'

Other studies have shown effects of render style, stylization, and importance of matching gender of actor to avatar:
e.g.,: 'Render me Real? Investigating the Effect of Render Style on the Perception of Animated Virtual Humans', 'Stylize or not to Stylize? Effect of Shape and Material Stylization on the Perception of Computer Generated Faces' , 'Evaluating the Effect of Emotion on Gender Recognition in Virtual Humans'

I would suggest to the authors to use just one avatar and ensure that the lighting is consistent and the gender of the avatar is matched to the actor for a fairer comparison of the motion techniques.

2. The sentences are super-short and are different across the different conditions. There is no way to determine if it is the sentence/word that is being performed or the avatar/motion conditions that are giving the results. I would suggest to the authors to chose a range of sentences and to apply them across all of the avatars for a fairer comparison.

In general, if the authors are going down the route of data-driven sign-language, I would suggest that finger motion is highly important and that they should be focusing on getting this part working well (focusing on words that have no finger movements is not going to work for many cases). Vision-based systems such as open pose are not really sufficient right now for accurate finger tracking. I would suggest to look at some of the available finger-capture gloves as a good solution for accurate finger tracking.

---

### Official Review · Reviewer_f4Fs · 2021-07-16
**Interesting and relevant topic, but quite preliminary work with from some methodological issues. Results are thus limited in informativeness and significance.**

**Rating:** 5
**Confidence:** 5

**Review:**

The paper address the interesting, quite old issue of how to synthesize sign language. Despite of a lot of approaches in the 90's, there is not much work on using modern ML techniques to approach this problem. The authors do not discuss why this is the case (do the rule-based, procedural systems perform sufficiently well?). The paper first presents experiences the authors have gained by applying a number of different tools and methods to video data in order to map video data directly to an avatar, using pose extraction or motion tracking. The experiences are reported in a rather colloquial style (e.g. reporting qualitatively and showing example images only), without a rigoros evaluation. Also, the descriptions are often at a low-level of detail and conceptual argumentation (e.g. it is not necessary to report that you were first planning and then rejecting to use NAO which is obviously not providing enough DoFs nor the required facial expressivity). This first part of the paper is hence not really informative. The second part  reports on more interesting user study comparing four different signers (three avatars, one human), which were taken from other researcher's work. However, there are severe methodological issues with the study. For example, the choice of measures (questionnaire) such as Godspeed or RoSAS seems not necessarily appropriate and is not well motivated. Likewise, the four avatars perform different behavior (signs) are thus not directly comparable.

---

### Official Review · Reviewer_1ko9 · 2021-07-18
**reject due to improper and meaningless evaluation, along with bad presentation, so no contribution can be found at all**

**Rating:** 4
**Confidence:** 5

**Review:**

The authors present an evaluation of a data-driven generative model for synthesis of manual gestures for the Kazakh-Russian Sign Language.

The document's language is not very good, with various English phrasing mistakes that result in many sentences being difficult to understand, and some even becoming meaningless. I highly recommend the authors to run their submissions through an English proofreading service before submitting as I consider this below the acceptable quality of English for a conference submission.

The topic of the paper is highly relevant for both the workshop, the ICMI/IVA community and even society in general. It is also a field with a clear lack of work so it is important to value such efforts.

The authors briefly describe their method from a very high level but do not clarify if it is an existing method, or a novel one, nor what is novel about it. There is also no note on what is the enhancement of the proposed method in comparison to existing and described methods such as MTC.

The major focus of the submission seems to be in the evaluation of the proposed technique and how it compares to the performance of an avatar animated using motion capture, and also a non-K-RSL knowing human performer. It is not clear what the difference is between Avatars 1 and 2, is it just the visual aspect? If so, what was the criteria to decide to use two, and why the characteristics of these two? Was there an intention to measure any difference between them?

It is mentioned that each avatar performs a different sentence and that the chosen sentences are very short and simple. I see two major problems with this, namely that such simple sentences do not seem enough to evaluate such a model like this one, and that if they are testing the variables of avatar appearance and performance methodology then they should do the best not to modify any other variable which might affect the evaluation, namely the performed sentence should be exactly the same across all different test cases. It is very difficult if not even very questionable to draw any conclusion from the study as it was presented.

The authors mention that the measurement instruments were parts of the Godspeed questionnaire, some custom Likert-scale questions, and four yes/no questions.
I was very disappointed to find that only the results from the Godspeed are presented, especially given that the paper is 6 pages long while the limit is 8 pages.

With such amount of remaining space it would also be relevant to include the additional custom questions given that there is no reference to consult for them and therefore they are absolutely unknown to the reader. If they are custom, they should be presented.

Finally, the use of the Godspeed questionnaire was also wrong, revealing that the authors did not follow the instructions on how to use the scales.
There are 5 dimensions in the Godspeed series: anthropomorphism, animacy, likeability, perceived intelligence and perceived safety. One may choose to only measure some of the dimensions, namely, the last one is especially formulated for robots and does not apply well to virtual agents. After one chooses the dimensions to measure, they must include all items within that dimension. The authors selected different items of different dimensions and measured them all individually, that is meaningless as it is not how the scale should be used. Each item within a dimension should be measured; then a scale reliability test such as the Cronbach's alpha should be used to first confirm if the measurement was reliable, and if reliability is confirmed, then the average of the items should be considered for the score of the dimension. The scores of individual items as presented have no scientific validity or meaning.

Unfortunately with all this there does not seem to be any chance to accept the paper into the workshop at the moment, so I hope the authors can find the opportunity to address the comments as soon as possible and to rerun and resubmit the study which will definitely become a valuable contribution.

---

### Decision · Program_Chairs · 2021-07-19

Reject